Small facial image dataset augmentation using conditional GANs based on incomplete edge feature input

Hung Shih-Kai sh19143@essex.ac.uk
Gan John Q.
School of Computer Science and Electronic Engineering, University of Essex , Colchester , United Kingdom
Srinivasan Kathiravan
Electronic publication date: 2021 Nov 17
Publication date: 2021
Volume: 7
Electronic Location ID: e760
Received 2021 Aug 24; Accepted 2021 Oct 6
Copyright: ©2021 Hung and Gan
Copyright year: 2021
Copyright holder: Hung and Gan
License: This is an open access article distributed under the terms of the Creative Commons Attribution License, which permits unrestricted use, distribution, reproduction and adaptation in any medium and for any purpose provided that it is properly attributed. For attribution, the original author(s), title, publication source (PeerJ Computer Science) and either DOI or URL of the article must be cited.
License URL: https://creativecommons.org/licenses/by/4.0/

Keywords: Generative adversarial networks, Deep convolutional neural networks, Image data augmentation, Small training data, Overfitting

Funding: The authors received no funding for this work.

==============================
Image data collection and labelling is costly or difficult in many real applications. Generating diverse and controllable images using conditional generative adversarial networks (GANs) for data augmentation from a small dataset is promising but challenging as deep convolutional neural networks need a large training dataset to achieve reasonable performance in general. However, unlabeled and incomplete features (e.g., unintegral edges, simplified lines, hand-drawn sketches, discontinuous geometry shapes, etc.) can be conveniently obtained through pre-processing the training images and can be used for image data augmentation. This paper proposes a conditional GAN framework for facial image augmentation using a very small training dataset and incomplete or modified edge features as conditional input for diversity. The proposed method defines a new domain or space for refining interim images to prevent overfitting caused by using a very small training dataset and enhance the tolerance of distortions caused by incomplete edge features, which effectively improves the quality of facial image augmentation with diversity. Experimental results have shown that the proposed method can generate high-quality images of good diversity when the GANs are trained using very sparse edges and a small number of training samples. Compared to the state-of-the-art edge-to-image translation methods that directly convert sparse edges to images, when using a small training dataset, the proposed conditional GAN framework can generate facial images with desirable diversity and acceptable distortions for dataset augmentation and significantly outperform the existing methods in terms of the quality of synthesised images, evaluated by Fréchet Inception Distance (FID) and Kernel Inception Distance (KID) scores.

Introduction

Deep convolutional neural networks generally require a large amount of training data to improve accuracy on unseen data or testing data (Tan et al., 2018). For many practical applications, it is expensive to collect a large amount of training data for deep learning. Data augmentation is often used to generate more training data. Traditional approaches for data augmentation include geometric transformations, such as translation, scaling, flip, rotation, etc., especially for image data (Mikolajczyk & Grochowski, 2018). However, the diversity introduced by traditional augmentation methods is limited and insufficient for many applications. The motivation of this work is to augment a small image dataset by making use of conditional edge features extracted from the available training images, and it can be expected that the synthesised images are more diverse and less distortive than those obtained from traditional methods.

In recent years, conditional generative adversarial networks (GANs), which can generate photorealistic images from conditional data, have become one of the most popular research fields in image synthesis. Conditional inputs, such as edges, mark points, masks, semantic maps, labels and so on, can be used to manipulate the images generated by GANs, making the synthesised images not only diverse but also controllable (Isola et al., 2017; Torrado et al., 2020). Image-to-image translation methods using conditional GANs directly learn the pixel mapping relationship between conditional edge features and real images (Lin et al., 2018; Wang & Gupta, 2016). Although image-to-image translation methods based on conditional GANs have been developed for controllable image synthesis, there are still several problems that should be resolved when applying them on a small training dataset: (1) Compared with unconditional GANs, a limitation of conditional GANs is that the output images must be generated from the corresponding conditional inputs, hence a clear mapping relationship between input and output should be correctly established. Corresponding mapping relationships are hard to be discovered, especially when only a very small training dataset is available for deep neural networks to learn. (2) With a small training dataset, the training process for image-to-image translation is easy to converge but difficult to obtain high-quality results due to the overfitting problem and insufficient information about the underlying data distribution, whether the used conditional features are of high quality or not (Dimitrakopoulos, Sfikas & Nikou, 2020), the discriminator will overfit the training data, and the generator would produce unexpected distortions in the generated images in the validation or application phase (Arjovsky & Bottou, 2017; Gu et al., 2019). (3) Training GANs using a small dataset must deal with the inevitable problem of mode collapse (Zhao, Cong & Carin, 2020), which implies that the GANs may learn the training data distribution from a limited number samples only but overlook other useful training data (Srivastava et al., 2017). Other issues, such as non-convergence and instability, would also worsen the quality of the generated images (Salimans et al., 2016). Clearly, it is a challenging task to synthesise photorealistic images using conditional GANs based on incomplete conditional features and a small number of training images.

Edge-based image-to-image translation using conditional GANs has the advantage of introducing diversity in image data augmentation, but it is challenging in terms of generating high-quality photorealistic images. Extracted edges cannot be regarded as perfect conditional features that support various advanced visual tasks and contain all visual information of potential perceptual relevance (Elder, 1999). Since edges generally contain incomplete information, such as unintegral geometry, simplified lines, discontinuous shapes, missing components, and undefined contours, it is hard for image-to-image translation methods to map edges to realistic images without clear conditional information.

To simply demonstrate the impact of the number of training images and incomplete conditional edges on the quality of the images generated using conditional GANs, some preliminary experimental results are shown in Fig. 1. It can be observed that the level of detail in the input edge features and the number of training images have considerable influence on the quality of both training and inference results, which is more serious on the inference results though because the input edge features used in the inference process have not been used in the training process.

Figure 1 Examples of training results (left side) and inference results (right side) with a different number of training images and different conditional edge inputs by using the same parameter setting for the image-to-image translation method.

Figure source credit: © Liu et al. (2015).

In this paper, a new image-to-image translation framework using conditional GANs is proposed, which can generate diverse photorealistic images from limited edge features after training with a small number of training images. Instead of deepening the convolutional layers or increasing the number of parameters, the proposed conditional GAN framework can learn additional relationships between incomplete edges and corresponding images, because regional binarisation and segmentation masks are used as new reference information, which can be obtained automatically by image processing. In particular, the proposed method can beneficially obtain extra pixel correlations between conditional edge inputs and the corresponding ground truth images to mitigate the influence of the overfitting problem during training. If the conditional GANs can efficiently learn from informative conditional inputs, such as colour, texture, edges, labels, etc., then it would be effortless to generate corresponding photorealistic image outputs (Dosovitskiy & Brox, 2016; Wei et al., 2018). A new network structure is proposed in this paper to divide the image synthesis task into two stages: the first stage transforms conditional input with incomplete edges into refined images as the new conditional input for the second stage, whose pixel values are obtained by combining the information from segmentation masks and binarised images. The second stage transforms the refined images into photorealistic image outputs.

The experimental results have demonstrated that the proposed method can generate high-quality diverse images even with a very small training dataset and very sparse edge features as conditional input. In addition, the generated images do not contain large distortions from incomplete or modified edge inputs for data augmentation purposes. The contributions of this paper are as follows:

• In order to deal with the problem of distortions in the images generated by conditional GANs due to using very small training data and sparse conditional input for diverse data augmentation, a new conditional GAN framework has been proposed, which converts the original incomplete edges into new conditional inputs in an interim domain for refining images and thus alleviates distortions caused by small training data and incomplete conditional edges.

• For the first time, the proposed method uses the mixture of pixel values of both binary images and segmentation masks to enhance the conditional input in an interim domain for refining images, which can integrate facial components, including eyes, nose, mouth, etc., so as to introduce diversity and enhance the quality of the images generated by conditional GANs trained using a very small training dataset.

• A facial image augmentation method using conditional GANs has been proposed, which can generate photorealistic facial images of diversity from incomplete edges or hand-drawn sketches. Compared with existing edge-to-image translation methods without ideal conditional inputs, the proposed method is tolerant to various incomplete edges as conditional inputs and able to generate diverse images of higher quality in terms of Fréchet Inception Distance (FID) (Heusel et al., 2017) and Kernel Inception Distance (KID) (Binkowski et al., 2018).

This paper is organised as follows: In ‘Related Work’, the related work is reviewed, including methods for image data augmentation, image-to-image translation, challenges in training GANs using small datasets. In ‘Methods’, the proposed methods, including the proposed conditional GAN framework, image pre-processing for generating the interim domain, and training strategies, are described in detail. ‘Experiments with the Proposed Conditional GAN Framework’ describes the experiment design, including data preparation and implementation details. The experimental results are presented and the performance of the proposed conditional GAN framework is evaluated qualitatively and quantitatively in ‘Results and Performance Evaluation’. Finally, conclusions, limitations and future work suggestions are presented in ‘Conclusions and Future Work’.

Related Work

The method proposed in this paper aims to use incomplete or modified edge features to augment small facial image datasets. The method was based on image-to-image translation using conditional GANs. This section introduces related techniques for image synthesis using conditional GANs.

Image data augmentation

Image data augmentation is applied in many applications to increase dataset size and data diversity. Deep neural networks are not easy to be trained well using small training datasets due to overfitting problems (Bartlett et al., 2020). One of the solutions to overcome overfitting is training data augmentation, which intends to boost the diversity of a small training dataset (Shorten & Khoshgoftaar, 2019). Traditional methods such as rotation, reflecting, translation, scaling, cropping, blurring, grey scaling and colour converting, have been commonly used for image data augmentation to reduce overfitting when training deep neural networks for image classification applications. Although these traditional techniques can produce similar images of high-quality, they rarely enlarge the feature diversity in the original images. Therefore, developing novel methods image data augumentation, which can synthesise not only diverse but also photorealistic images based on a small image dataset, is a very important but challenging task.

Many GANs have been proposed for image synthesis (Gatys, Ecker & Bethge, 2016; Iizuka, Simo-Serra & Ishikawa, 2017). Since conditional GANs provide alternative methods to edit images as well as manipulate generative attributes, they have been applied to generate high-quality images of diverse features (Lin et al., 2018). Image synthesis using conditional GANs can boost the applications of data augmentation in many real applications, but it is hard to generate high-quality diverse images when the training dataset is small.

Image-to-image translation and image synthesis using conditional GANs

Image-to-image translation is a type of image synthesis using conditional GANs with specific forms of conditions, such as videos and images (Azadi et al., 2018; Szeto et al., 2021), scenes (Ashual & Wolf, 2019; Johnson, Gupta & Li, 2018), or segmentation masks (Cherian & Sullivan, 2018; Park et al., 2019). The conditional inputs can be transferred from a source domain to a target domain by using supervised learning techniques. The main concept of image-to-image translation is to learn the data mapping relationships (Liu, Breuel & Kautz, 2017). Image-to-image translation methods can automatically generate images between the corresponding domains (Mo, Cho & Shin, 2019) and discover the dependence with pairs of images to translate features into realistic images. Image-to-image translation methods provide a prominent approach to image synthesis with diverse results by using controllable conditional features (Isola et al., 2017; Zhu et al., 2017).

With remarkable advantages in image-to-image translation, edge-to-image synthesis has achieved visually pleasing performance (Yi et al., 2017). Compared with other conditional forms, edges are one of the most easy-to-obtain and simple features in computer vision, and hand-drawn sketches can be regarded as a specific form of edge features (Chen & Hays, 2018). Since edges usually contain critical information, such as gradients, shapes, contours, profiles, boundaries and so on, they directly provide simple and direct depictions for objects (Chen et al., 2009; Eitz et al., 2011). The benefit of using edge features is that they are flexible to modify for introducing diversity in data augmentation. However, in contrast to labelled images, edges as conditional inputs often preserve insufficient information (e.g., textures, colours, brightness, labels, etc.), which makes it hard to generate desirable high-quality images using edge-to-image translation.

Face synthesis requires integral contours that faithfully reflect the inputs and connections to the realistic context (Li et al., 2020). Incomplete features with missing parts or undefined components will affect the quality of images generated by conditional GANs (Jo & Park, 2019; Karras et al., 2018). To alleviate the influence of lacking ideal conditional inputs, this work proposes to reconstruct the missed features in the conditional input and transform limited edge information into refined images by introducing an interim domain to alleviate the negative effect of imperfect conditional features on the quality of the generated images.

Challenges in training GANs using small training dataset

Mode collapse is a major problem in training GANs using small training data, which makes it fail to switch training samples and thus the trained GANs would be over-optimised with a limited number of training samples only (Odena, Olah & Shlens, 2017). Due to the mode collapse problem, the generator is constructed with very limited training data, which makes the discriminator believe that the generative outputs are real instead of fake. This is one of the key causes that make it difficult for GANs to generate diverse results for data augmentation purposes (De Cao & Kipf, 2018).

Another drawback in training GANs using a small training dataset is that it is hard to fine-tune all parameters to discover an optimal balance between the generator and discriminator. Both the generator and discriminator contain a large number of trainable parameters, which necessarily need enormous training data to prevent from overfitting problems (Gulrajani et al., 2017; Mescheder, Geiger & Nowozin, 2018). Ideally, training a GAN requires a large amount of training data for optimising its parameters and reducing losses to generate photorealistic image outputs (Wang et al., 2018c). Therefore, reducing the requirement for a large amount of training data is a grand challenge in using GANs for image data augmentation in which available training data is limited. Our previous preliminary work (Hung & Gan, 2021) partly addressed the above challenges by proposing a new conditional GAN architecture. This paper substantially extends our preliminary work via further investigation and deeper analysis of much more experimental results.

Methods

Image-to-image translation methods find specific mapping relationships between source distribution and target distribution. In general, a small number of paired features may not comprehensively align with the source and target distributions using imperfect conditional inputs such as incomplete edges and a small training image dataset. Therefore, data refining in paired features can be adopted to expand mapping relationships based on a small training dataset. In this section, a new method is proposed for facial image synthesis based on a very small training image dataset.

With incomplete conditional features in the source domain and small training data in the target domain, the mapping between source and target domains cannot be described by clear one-to-one relationships. The method proposed in this paper transfers the source domain to an interim domain for refining images with extra annotated information, in which newly defined images in the interim domain need to be generated based on a small training dataset. The interim domain can provide extra reference information to discover more mapping relationships between the source and target domains.

Figure 2 shows the proposed translation method using a small training dataset. It is difficult using a small training dataset to obtain a comprehensive view of correct mapping relationships between the source domain and target domain without sufficient representative training samples, as shown by the blue line in Fig. 2. Even if changing the types of the conditional inputs, a similar situation remains as it is still difficult to learn correct mapping relationships, as demonstrated by the red line in Fig. 2. For the purpose of comprehensively finding correct mapping relationships, extending the mapping relationships in an interim domain for refining images, as shown by the green dotted line in Fig. 2, can reduce uncertainty caused by using a small training dataset and incomplete edge features as conditional inputs for data augmentation with diversity. This will be explained in more detail when introducing the proposed conditional GAN framework later.

Figure 2 The proposed translation method by defining an interim domain for refining images based on a small training dataset.

Image pre-processing is adopted in the proposed conditional GAN to enhance the mapping relationship from source domain to target domain. Figure source credit: © Liu et al. (2015).

When training GANs using small training datasets, the following factors should be considered: (1) It is difficult to avoid distortions in the generated images and training imbalance with a small number of training images or insufficient diverse samples. (2) Through deep convolutional neural network structures, such as convolution, normalisation and downsampling, it is easy to lose spatial information and impractical to completely preserve the conditional information with a small number of training images (Kinoshita & Kiya, 2020; Wang et al., 2018b). If the conditional inputs contain sparse, unclear, limited, discontinuous, or incomplete features, fine-tuning model parameters without distortions becomes much more difficult. (3) Using a small training dataset and limited conditional features will make the training easy to overfit but hard to obtain realistic results. Since many parameters in a deep convolutional neural network need to be fine-tuned, it is impossible to optimise all the parameter values using a small training dataset in terms of the generalisation ability of the trained deep neural network. To tackle the above problems in training GANs using a small number of training images, several training strategies are adopted in the proposed method, which are described as follows:

Enlarging the diversity of source domain: The training of a GAN using a small training dataset can easily converge but frequently attain unrealistic inference results, mainly because of the overfitting problem. It is impossible for GANs to have a whole view of the target domain through training with a limited number of training images. For the goal of achieving photorealistic results, both the discriminator and generator should stay in an equilibrium balance. Increasing the data diversity and widening the mapping relationship between the source domain and target domain could help achieve the required balance between the generator and discriminator when using a small number of training images. In the proposed method, new reference information is created by image pre-processing, and the adoption of the interim domain for refining images can enlarge the diversity of the source domain.

Double translation: Double translation strategy aims to decrease the chance of mode collapse in single translation approach and reduce the impact of the uncertainty due to using incomplete edges as conditional input, so as to alleviate the distortions caused by training with a small number of training images. For generating additional reference information, the proposed method combines binary images and segmentation masks to generate refined images as conditional input in the next translation. In the first translation, refined images with annotated facial components are generated from incomplete edge features. This translation is conducted between the source domain and the interim domain. The second translation is conducted between the interim domain and target domain, which can successively learn from the possible distortions in the first stage to avoid or alleviate negative distortions in the final outputs.

Reusing the conditional information: Spatial information in the conditional edges can be easily lost during training in the convolutional neural layers, and the relationships between the source domain and target domain will become incomprehensive. In order to reduce the spatial information vanishing, edge features in the source domain can be reused in each translation.

Freezing weights: Weight freezing is a strategy to overcome the gradient vanishing problem during training, which often happens when using a small training dataset. If the provided training data cannot give the discriminator enough information to progress the generator, the gradient will become smaller and smaller when going from bottom to top layers of the network. Incomplete conditions would worsen the gradient vanishing problem and make it impossible to fine-tune the model parameters to obtain realistic results. Hence, freezing part of weights in separate training stages allows the discriminator to acquire information from each training stage rather than tuning all parameters at one time.

The proposed conditional GAN framework

To mitigate the output distortions caused by using a small training dataset and incomplete edge as conditional input, additional paired segmentation masks and regional binary images are used as reference information in the proposed method, which can enrich the mapping relationships between the source domain and target domain. Consequently, the proposed method creates additional data distributions from the small training dataset using image pre-processing, and the data in the interim domain provides more referable features than the original incomplete edges in the source domain.

Two U-nets (Ibtehaz & Rahman, 2020; Ronneberger, Fischer & Brox, 2015) are adopted in the proposed conditional GAN framework for image-to-image translation. This structure can achieve better performance when training conditional GANs using small training data for two reasons: on one hand, during the training process the U-nets create images based on the special concatenating structure, which is beneficial to retain the matched features from limited conditional features for integral perceptions in convolutional layers. On the other hand, the U-net structure is simple and beneficial to generate images without using very deep convolutional layers, which is critical to alleviate the gradient vanishing problem during training using small training data and incomplete edges. The proposed framework also reuses the conditional input information to strengthen the input features at each training stage, and freezing weights for separate networks at each training stage can prevent from gradient vanishing as well. To sum up, the proposed conditional GAN framework can alleviate the problems in training GANs using small training data by intensifying the conditional information in the source domain. An overview of the proposed conditional GAN framework is shown in Fig. 3 and described as follows.

Figure 3 Overview of the proposed conditional GAN for translating edges to photorealistic images using two U-nets.

Figure source credit: © Liu et al. (2015).

The proposed model consists of three parts: (1) image pre-processing, (2) generators and (3) discriminators. The two generators use the same convolutional structure of the U-net, both of which down-sample and then up-sample to the original size of input images. All convolutional layers use convolution kernels of size 3  × 3 (Yu et al., 2019), and normalisation is applied to all convolutional layers except for the input and output layers (Zhou & Yang, 2019). In the training phase, the first generator is used to create refined images based on the original sparse edges and ground truth. The refined images are referred from image pre-processing, which contain features related to texture, colour, shape of different facial components. The second generator is designed to improve the synthetic process to generate photorealistic images from the interim domain. In the inference phase, the generators use the fine-tuned parameters to generate photorealistic images from conditional edges that may have not been seen during training. The two discriminators have the same task of distinguishing between real and fake images: the first one is to identify generated images in terms of refined images, and the other is in terms of the ground truth.

Image pre-processing and refining

Image refining is essential for providing informative conditional features since incomplete edges may contain much unidentical information representing the same facial component. This uncertainty makes it difficult for conditional GANs to comprehensively find pixel relevance in different domains. For instance, an unclear “black circle” with incomplete edges can represent either nose, ear or eye, even if using a powerful network, it is difficult to learn well with a rare sign of “circle” as conditional input without any other crucial information (e.g., colour, types, angels, positions, textures, sub-components, brightness, layouts, shapes, etc.). A refining process can be employed within an image-to-image translation method, which enhances powerful one-to-one mapping by providing close to ideal conditional input. However, there is no guarantee that ideal conditional inputs can be obtained in real applications, especially if the conditional inputs are incomplete or sparse edges. These uncertainties can result in unexpected distortions. If the interim domain for refining images can provide more specific information, the synthetic quality will be improved. Therefore, enhancing conditional information is one of the important goals of image pre-processing and refining.

Edge extraction

Edges may contain incomplete features with many possible feature types, including undefined density, shape or geometry (Royer et al., 2017). However, to achieve high performance, image-to-image translation methods need clear conditions (Lee et al., 2019). In order to generate photorealistic images from limited conditional information, extending translation relationships based on proper reference images can make the mapping relationships between the source domain and target domain more precise based on a small training dataset. As an example, the corresponding mapping relationships among the ground truth, conditional features, and refined image are shown in Fig. 4. The ground truth image is responsible for providing not only realistic features but also reference images to composite refined images. The red boxes shown in Fig. 4 indicate the eye mapping in different domains, and the new relationships are expected to effectively reduce the uncertain mapping in image-to-image translation.

Figure 4 Corresponding mapping relationships among the conditional inputs, refined image and ground truth.

Figure source credit: © Liu et al. (2015).

Adoption of an interim domain

In contrast to directly transforming the source edges to target results, the proposed conditional GAN framework converts conditional edges to refined images in an interim domain first. The interim domain trends to reconstruct possible missing information from incomplete conditional features using a U-net. Mode collapse problem may happen in the interim domain when incomplete edges are transferred to a refined image. Nevertheless, the translation at this stage is useful for facial component identification because the incomplete edges in the source domain are further processed. The refined images provide clearer accessorial information than the original incomplete edges, even if they are converted into simplified samples when mode collapse happens. By trial and error, appropriate regional features as reference images can reduce distortions and mismatch of features based on very limited edges. Therefore, the refined images are constructed by combining binarised images and segmentation masks, as shown in Fig. 3. In short, the main function of the interim domain is to refine the original data distribution and strengthen the incomplete conditional features from the source domain.

Figure 5A shows inference results of using uncertain edges to generate segmentation masks from 50 paired training images. To handle the incomplete edges as the conditional input, facial components can be reconstructed by a U-net in the interim domain. It is evident that the proposed conditional GAN can learn from only 50 segmentation masks to generate more integral face components, such as nose, eyebrow, hair and mouth. Figure 5B illustrates examples where incorrect eye shapes are obtained, as shown in the red boxes, which would aggravate distortions in the target domain. What is worse, this situation is hard to be solved because it is difficult to increase the number of diverse samples based on a small dataset as GANs generally require more diverse data to be trained well. To resolve this problem, additional binary images with clear regional information are obtained through image pre-processing, which can enhance the contours and thus reduce distortions in facial components, as shown in Fig. 4. In contrast to imprecisely depicting facial components in segmentation masks, binary images obtained by appropriate thresholding can produce more correct contours than segmentation masks and thus alleviate the problems caused by very limited training data.

Figure 5 Inference results in translating sparse edges to labelled segmentation masks with 50 training images.

(A) The outputs can roughly resume the missing facial components from incomplete layouts when given abstract inputs. (B) The red boxes indicate the corresponding indefinite contours in the original inputs and generated masks. Figure source credit: © Liu et al. (2015).

Figure 6 shows that binary images can handle uncertain edge density in the inference phase to enhance crucial edge information with regional distributions. Binarised regional features can be extracted by the corresponding edge distribution from a small training dataset, which can not only integrate crucial contours, as shown in Fig. 6A, but also get rid of meaningless noise when various untrained edges may be unrecognisable in the inference phase, as shown in Fig. 6B. It is noteworthy that the results presented in Figs. 5 and 6 can be regarded as those from an ablation study, which shows that removing the component of combining binarised regional features in the proposed method will significantly deteriorated the performance of the proposed conditional GAN.

Figure 6 Inference results in translating sparse edges to binary regional images with 50 training images.

(A) The outputs can integrate discontinued contours when given sparse inputs. (B) The outputs can get rid of ‘bogus’ edges when given very dense inputs. Figure source credit: © Liu et al. (2015).

Model training and loss functions

Conditional adversarial loss

During training the proposed conditional GAN framework, it is difficult to find a balance between the generator and discriminator, especially when only very limited training data is available. Using an appropriate loss function is critical to ensure good quality of the generated images. Firstly, to distinguish real images from fake ones, the following basic loss function is used for the two convolutional neural networks, which is known as conditional adversarial loss. LadvD,G=EI,SlogDS|I+EI,I′log1−DI,GI′|I

where 𝔼 represents expected value, G the generator, D the discriminator, S the source image, I the conditional edge feature input, and I’ the generated image. In the first U-net, S should contain a mixture of pixels of binary image, segmentation mask and ground truth so as to distinguish between real refined image and fake generated image. In the second U-net, S needs to be set as the ground truth only.

Feature matching loss

As in the pix2pix GAN model (Isola et al., 2017), the L 1 pixel loss as feature matching loss in the synthesised images is adopted. Since there are paired images in the training phase, the L 1 distance between the generated image (I′) and source image (S) can be defined as follows: LL1G=ES,I,I′S−GI′|I1

Overall loss

The main purpose of using the loss function is to help the generator to synthesise photorealistic images by minimising the loss value with a limited number of input images. The overall loss function is defined as minGmaxDLadvD,G+αLL1G

where α is a weight parameter. A larger value of α encourages the generator to synthesise images less blurring in terms of L1 distance.

The second U-net uses the refined images and original sparse edges as inputs to generate photorealistic images with the same loss function but different training parameters and freezing weights. Another difference between these two networks is the source image S, which should be either the refined images or the ground truth images.

Experiments with the proposed conditional GAN framework

Data preparation

A small set of randomly chosen images from CelebA-HD (Liu et al., 2015) formed the training image dataset in our experiments. CelebA-HD includes 30,000 high-resolution celebrity facial images. All the images were resized to 256  × 256 for our proposed model. CelebAMask-HQ (Lee et al., 2020) is a face image dataset consisting of 30,000 high-resolution face images of size 512  × 512 and 19 classes, including skin, nose, eyes, eyebrows, ears, mouth, lip, hair, hat, eyeglass, earring, necklace, neck, cloth and so on. All the images in CelebAMask-HQ were selected from the CelebA-HD dataset, and each image has segmentation masks of facial attributes corresponding to CelebA-HD.

Since different numbers of segmentation masks were used to compare the performance of different methods with different numbers of training samples, the CelebAMask-HQ was used as the standard segmentation masks of reference images. If a very small training dataset is used, it would be fine to manually generate the segmentation masks by image pre-processing. In our experiments, the segmentation masks from CelebAMask-HQ were used as the common reference images of the corresponding training images.

Implementation details

The hyper-parameter values were determined through trial and error in our experiments. For training the proposed conditional GAN framework, the Adam optimiser was used to minimise the loss function with the initial learning rate set to 0.0002 and the momentum 0.5. The weight parameter α in the loss function was set to 100. All the experiments were conducted on a desktop computer with NVIDIA GeForce RTX 2080 GPU, Intel Core i7-6700 (3.4 GHz) processor, and 16G RAM.

Incomplete edges or hand-drawn sketches as conditional inputs usually represent abstract concepts, which are beneficial for generating diverse data augmentation results but it is difficult for conditional GANs to generate photorealistic images with limited conditional inputs based on small training data. In our experiments, edges were extracted by Canny edge detector (Canny, 1986), which can obtain simple and continuous edges using a set of intensity gradients from realistic images. Two intensity gradient magnitudes are used in Canny edge detector as a threshold range to control the edge density, which is determined in our experiments by a threshold ratio. It is the ratio of the high threshold value to the maximum magnitude, and the low threshold value is 40% of the high threshold value. The edges produced by the Canny edge detector are more similar to hand-drawn sketches than those by other commonly used edge detectors, as shown in Fig. 7. The appropriate threshold ratio for the Canny edge detector was chosen through trial and error in our experiment. The red box in Fig. 7 shows the edges extracted using the chosen threshold ratio, which has clear information about facial components without unexpected noise and meets the requirement for good conditional inputs.

Figure 7 Comparison of different edge detectors: (A) Results of Canny. (B) Results of Sobel. (C) Results of Laplace. (D) Results of Gradient.

Figure source credit: © Liu et al. (2015).

In the design of the interim domain, the pixel values of the refined image were set by the following mixture ratios: 25% from binary image, 25% from segmentation mask, and 50% from original image. Figure 8 shows the inference results of the refined images and the corresponding generated image outputs. The red boxes represent blending areas in the masks, binary images and texture features in the refined images, which reflect the brightness changes in the generated image outputs. The overlapped regions are visually darker and gloomier compared to other regions. Therefore, these blending areas from different reference images conduct transitions in brightness and lightness to synthesise realistic results. With the interim domain, the proposed conditional GAN can efficiently deal with both overlapped and non-overlapped mappings between segmentation masks and binary regions, leading to more photorealistic image outputs.

Figure 8 Inference results with examples of refined images and final outputs.

The red boxes represent blending areas in the refined region, which can be reflected by the brightness in the generated image outputs. Figure source credit: © Liu et al. (2015).

Image blending with different styles is beneficial to augment training image datasets with diversity. In the proposed conditional GAN framework, generated images were controlled by conditional edge inputs. Exchanging or modifying edge features is an easy way to generate different images that increase data diversity and expand original facial features. Figure 9 shows examples of the generated images with facial features swapped on a small training dataset by exchanging edge components in conditional inputs. It can be seen that the generated images can preserve facial features with clear conditional edges and reconstruct the critical components in incomplete or undefined areas.

Figure 9 Results from exchanging conditional facial edges to generate diverse styles of facial images.

Figure source credit: © Liu et al. (2015).

Results and performance evaluation

For performance evaluation, the proposed conditional GAN framework was used to generate images from different training images and conditional edge input settings. To demonstrate the performance of the proposed method, the state-of-the-art edge-to-image translation methods were compared both qualitatively by visual inspection of the quality and diversity of the generated images and quantitatively in terms of FID and KID scores.

Diversity in facial image augmentation using the proposed conditional GAN

It is clear that the threshold ratio chosen for the Canny edge detector affects the density level of the extracted edges, which as conditional inputs would affect the quality of the images generated by the conditional GAN. It is desirable that the proposed conditional GAN can generate diverse images with the change of edge density levels in the conditional input but be robust in terms of the quality of the generated images. Figure 10 shows the inference results with different density levels in the conditional edges, which were not included in the training phase except for those in the red box. It can be seen that the generated images are slightly different with different density levels in the conditional edge inputs and the distortions are small even when the GANs were trained using a small dataset of 50 training images. The generated images are more photorealistic when the conditional input contains less noise or unidentical edges, which correspond to those generated with the edge density level chosen in the training phase, as shown in the red box. Fortunately, with the change of the density levels of the conditional edge inputs, the quality of the generated images is prevented from considerable deterioration because the refined images can integrally represent facial features at an acceptable level based on a small dataset. Consequently, as the conditional inputs to the second U-net in the second stage, they play an important role in reducing distortions in the generated facial image outputs.

Figure 10 Inference results with example images in the source, interim, and target domains respectively.

The various density levels in the conditional inputs were not included in the training phase except the one in the red box generated by Canny edge detector with threshold ratio = 0.4. The results are from GANs trained using 50 images only. Figure source credit: © Liu et al. (2015).

Figure 11 shows examples of facial image augmentation results using 50 training images to train the proposed conditional GAN framework. Diverse new facial images can be generated from each training image with extracted edges modified for desirable facial features as conditional inputs. The modifications to the extracted edges include adding or deleting parts of the edges or changing facial expression or direction, as shown in Fig. 11. It can be seen that the image data augmentation results using the proposed conditional GAN are more diverse than traditional augmentation methods and the generated images are of good quality due to the use of the interim domain. For data augmentation purposes, using deliberately modified edges as conditional inputs to the proposed conditional GAN framework can boost the data diversity on the basis of the available small set of training images.

Figure 11 Examples of facial image augmentation results using 50 training images, with parts of input edges modified for introducing diversity so as to augment each training image with desirable facial features.

Figure source credit: © Liu et al. (2015).

Figure 12 shows some other examples of facial image data augmentation using 50 training images to train the proposed conditional GAN, with edge features from multiple training images swapped as conditional inputs. The red boxes in the figure indicate the original training images, and the other images in a row are generated by the proposed conditional GAN, which shows swapped facial features (including eyes, eyebrows, nose and mouth) or hairstyles. It can be seen that by using mixed edge features from multiple training images as conditional inputs the proposed conditional GAN can efficiently generate diverse facial images of good quality.

Figure 12 Examples of facial image augmentation results using 50 training images, with face components and hairstyles in different training images swapped in the edges as conditional inputs so as to generate diverse facial images.

Figure source credit: © Liu et al. (2015).

In general, it is difficult for image-to-image translation methods to generate high-quality images with conditional inputs that are not directly corresponding to features in the training images, such as hand-drawn sketches. In the previous experiments, it has been demonstrated that the interim domain is helpful to generate high-quality images with various edge density levels for diversity. In our experiments, hand-drawn sketches were also used as conditional inputs for the proposed conditional GAN to generate facial images with specific facial expressions or features. Figure 13 shows the inference results with hand-drawn sketches as conditional inputs, with the proposed conditional GAN trained using a dataset of 50 training images. It is obvious that, when giving unidentical or incomplete facial contours in the conditional inputs, the refined images generated by the first U-net in the proposed conditional GAN structure are responsible for reducing distortions in the generated images while keeping the diverse facial expressions introduced by the hand-drawn sketches.

Figure 13 Inference results with example images in the source, interim, and target domains respectively.

The conditional inputs are hand-drawn sketches showing different facial expressions. The proposed conditional GAN was trained using 50 training images. Figure source credit: © Liu et al. (2015).

Qualitative comparison

To evaluate the quality of the images generated by the proposed conditional GAN framework, the images generated by the proposed method were compared with those generated by the state-of-the-art edge-to-image translation methods, including pix2pix (Isola et al., 2017) and pix2pixHD (Wang et al., 2018a), under the same training conditions and in terms how the generated images are comparable to the ground truth images. Figure 14 shows representative images generated respectively by the three conditional GANs for comparison, trained using the same small dataset of 50 training images. Different sparse edges as conditional inputs were tested. The results in Fig. 14 demonstrate that the method proposed in this paper can generate more photorealistic facial images with fewer distortions than pix2pix and pix2pixHD, when the GANs were trained using small training data.

Figure 14 Inference results generated with sparse edge inputs (the first row), in comparison with those obtained from the state-of-the-art conditional GANs.

The images were generated respectively by the three GANs for comparison, trained using the same small dataset of 50 training images. Figure source credit: © Liu et al. (2015).

Quantitative comparison

Since the number of training images is small, the difference between the generated images and the corresponding ground truth images is noticeable by visual inspection. In order to quantitatively compare the quality of the images generated by different conditional GANs, FID and KID scores were adopted to evaluate the photorealistic scales of the generated images in our further experiments. FID is widely adopted to evaluate the visual quality of generated images, which calculates the Wasserstein distance between the generated images and the corresponding ground truth images. KID can be used similarly for image quality measurement, but KID scores are based on an unbiased estimator with a cubic kernel (Binkowski et al., 2018). Clearly, lower FID and KID scores represent a better match between the generated images and the corresponding ground truth images.

To evaluate the effect of the interim domain adopted in the proposed conditional GAN framework on the quality of the generated images, the performance of double U-nets is compared with that of a single U-net in terms of FID and KID with different threshold ratios used in the Canny edge detector. In the training phase, one input type with threshold ratio = 0.4 and three input types with threshold ratios = 0.2, 0.4 and 0.6 were considered and 50 training images were used, whilst in the inference phase 11 different threshold ratios (0.01, 0.05, 0.1, 0.2, 0.3, 0.4, 0.5, 0.6, 0.7, 0.8 and 0.9) were tested with 1000 generated images respectively. Figure 15 shows the comparative results in terms of FID and KID scores. Three points can be made from the experimental results presented in this figure: Firstly, double U-nets achieved lower FID and KID scores than single U-net, indicating that the interim domain for refining images in the proposed conditional GAN can reduce distortions caused by incomplete conditional edges and small training dataset, and thus improve the quality of the generated images; Secondly, training with three conditional edge density levels, compared with only one conditional type, can achieve better and robust performance with various levels of conditional edge density (for better diversity); Thirdly, the most photorealistic performance was achieved when the edge density levels in the inference phase are close to those used in the training phase. The results presented in Fig. 15 can be interpreted from the perspective of ablation study because removing the interim domain adopted in the proposed method will considerably deteriorate the performance of the conditional GAN.

Figure 15 FID and KID scores of double U-nets with an interim domain and single U-net with different levels of input edge density, respectively.

One input type (threshold ratio = 0.4) and three input types (threshold ratios = 0.2, 0.4, 0.6) in the source domain were used respectively during training with a small training dataset of 50 images. The FID and KID scores were calculated based on the same 1000 inference images at different edge density levels.

This paper aims to generate photorealistic facial images using conditional GANs trained with a small set of training images for data augmentation. To evaluate the effect of the number of training images on the quality of the images generated by the proposed conditional GAN, pix2pix and pix2pixHD, different numbers (25, 50, 100, and 500) of training images were used to train each of the three conditional GANs separately. Moreover, in order to demonstrate the effect of different conditional edge density levels, both sparse edges (threshold ratio = 0.4) and dense edges (threshold ratio = 0.2) were used to generate 1000 images by each trained conditional GAN. The FID and KID scores of the images generated by the three conditional GANs were calculated respectively. Figure 16 shows the changes of FID and KID scores with the different numbers of training images, from which the following three points can be made: (1) With the interim domain, the proposed conditional GAN achieved lower FID and KID scores than pix2pix and pix2pixHD when trained with the same number of training images. (2) Dense conditional edges achieved lower KID and FID scores than sparse edges, but the diversity in the generated images may be constrained. (3) With the increase in the number of training images, the advantage of the proposed method over the existing methods becomes less obvious. This tendency indicates that the proposed conditional GAN framework is very effective when it is trained with a small number training samples and its performance would approach to that of the existing methods when the number of training samples becomes relatively large.

Figure 16 Changes of FID scores (first row) and KID scores (second row) with different number of training images.

Comparison among three edge-to-image translation methods with sparse and dense edge inputs respectively: pix2pix, pix2pixHD and ours.

Conclusions and Future Work

In this paper, a new conditional GAN framework is proposed for edge-to-image translation based on a small set of training data, which can synthesise photorealistic diverse facial images using incomplete edges as conditional inputs for data augmentation purposes. In order to solve the problem in training conditional GANs using small training data, an interim domain for refining images is introduced in the proposed conditional GAN, which can effectively reduce unexpected distortions and thus improve the quality of the generated images. Experimental results have demonstrated that blending segmentation masks and regional binary images as refined reference images can reduce distortions in facial components of the images generated by the conditional GAN trained with a small training dataset. Compared with the existing edge-to-image translation methods, the proposed conditional GAN can not only automatically transfer incomplete conditional edges to reference images with more facial features in the interim domain but also effectively reduce unexpected distortions caused by small training data.

Compared to directly transferring source domain into target domain, the proposed method can have a more comprehensive view to generate more photorealistic edge-to-image translation results when using various incomplete conditional edges for data augmentation. More informative reference images can be constructed in the interim domain from incomplete edge inputs to integrate useful facial components. The proposed conditional GAN trained using a small dataset can synthesise various photorealistic facial images by manipulating conditional edge features or using hand-drawn facial sketches for diverse image data augmentation. Compared to the existing conditional GANs for image-to-image translation, the images generated by the proposed conditional GAN have less distortion and more diversity, which is desirable for data augmentation purposes.

Due to limited GPU computing facilities available for conducting our experiments, it is hard to optimise the hyperparameters of the tested models, and the performance evaluation is based on the comparison with two state-of-the-art methods only. More extensive comparative study would be desirable in the future research to draw more reliable conclusions.

The advantage of the proposed conditional GAN framework over the existing methods becomes less obvious when the number of training samples are relatively large. For future work, the interim domain could be improved so that the proposed conditional GAN framework would also significantly outperform existing methods for image data augmentation when a reasonably large number of training image dataset is available.

Supplemental Information

Supplemental Information 1 Code

Click here for additional data file.

Supplemental Information 2 Augmented facial images with hand drawing lines

Click here for additional data file.

Supplemental Information 3 Augmented facial images with blending features

Click here for additional data file.

Supplemental Information 4 Augmented facial images with the Canny edge parameters of 0.1, 0.2 and 0.3

Click here for additional data file.

Supplemental Information 5 Augmented facial images with the Canny edge parameters of 0.4, 0.5 and 0.6

Click here for additional data file.

Additional Information and Declarations

Competing Interests

Author Contributions

Data Availability

The authors declare there are no competing interests.

Shih-Kai Hung conceived and designed the experiments, performed the experiments, analyzed the data, performed the computation work, prepared figures and/or tables, and approved the final draft.

John Q. Gan conceived and designed the experiments, performed the experiments, prepared figures and/or tables, authored or reviewed drafts of the paper, and approved the final draft.

The following information was supplied regarding data availability:

The CelebA dataset is available at: https://mmlab.ie.cuhk.edu.hk/projects/CelebA.html.

The CelebAMask-HQ dataset is available at GitHub: https://github.com/switchablenorms/CelebAMask-HQ.

The raw measurements and source code are available in the Supplementary Files.

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
