# Peer review of "Small facial image dataset augmentation using conditional GANs based on incomplete edge feature input"

_PeerJ Computer Science, doi:10.7717/peerj-cs.760_

## Round 0.1 · original submission · Minor Revisions

Kindly address the comments suggested.

·

Basic reporting

In this paper, a new image-to-image translation framework using conditional GANs is proposed, which can generate diverse photorealistic images from limited edge features after training with a small number of training images.The article has a certain degree of innovation and novelty, the experiment is relatively sufficient, and the overall content is complete, but there are still some problems that the author hopes to correct:
1)Whether the pictures involved in the article can be directly reflected in the text to facilitate reading and understanding;
2)The quality of the picture is not very high, it is recommended to replace it, and some of the picture content is redundant, and it is recommended to delete it;
3)Figure 16 compares the KID and FID values of different methods within 500 sheets, and the difference between the results of the other two methods and the method in this article is getting smaller and smaller. Can the author compare more images, or briefly Explain the change trend of subsequent FID and KID results;

Experimental design

no comment

Validity of the findings

no comment

Additional comments

no comment

Reviewer 2 ·

Basic reporting

It is better to include a paragraph providing details about the structure of the paper (could be the last paragraph of the Introduction section).

The related work section is very comprehensive (good work!). Include some details of additional image data augmentation works would further improve the related work section.

Indicate potential future works of this research in the conclusion section.

Experimental design

Structuring the methods section is very important to enhance the readability of the paper. It is better to include individual subsections to three major parts of the proposed method.

Lack of supportive arguments is the primary concern in the results section. Authors will get benefitted by including a comprehensive discussion on both qualitative and quantitative results.

An ablative study is a must in this case to express the significance of each module.

Validity of the findings

Highlight the limitations of the work is important for researchers in the field to explore more in this domain of interest. Discuss some of them in the results section.

Reviewer 3 ·

Basic reporting

There contains grammatical errors and typos in the manuscript. The authors should re-check and revise carefully.

Experimental design

The research question is well defined. Methods are described in full detail.

Validity of the findings

The findings are valid, and can be easily replicated.

---

## Round 0.2 · accepted · Accept

The manuscript requires careful proofreading to correct the typos.

·

Basic reporting

Great improvement. The article is well written, well structure and I believe the level of novelty is good. I really don't see anything especially to comment and I rather accept it within the current form.

Experimental design

no comment

Validity of the findings

no comment

Additional comments

no comment

Reviewer 2 ·

Basic reporting

The authors have carefully revised the manuscript according to previous reviewers' feedback.

Experimental design

Improved to the expected quality.

Validity of the findings

Improved to the expected quality.